# DO GANS LEARN THE DISTRIBUTION? SOME THEORY AND EMPIRICS

**Sanjeev Arora**
Department of Computer Science
Princeton University
Princeton, NJ 08544, USA
`arora@cs.princeton.edu`

**Andrej Risteski**
Applied Mathematics Department and IDSS
Massachusetts Institute of Technology
Cambridge, MA 02139, USA
`risteski@mit.edu`

**Yi Zhang**
Department of Computer Science
Princeton University
Princeton, NJ 08544, USA
`y.zhang@cs.princeton.edu`

## ABSTRACT

Do GANS (Generative Adversarial Nets) actually learn the target distribution? The foundational paper of Goodfellow et al. (2014) suggested they do, if they were given "sufficiently large" deep nets, sample size, and computation time. A recent theoretical analysis in Arora et al. (2017) raised doubts whether the same holds when discriminator has bounded size. It showed that the training objective can approach its optimum value even if the generated distribution has very low support —in other words, the training objective is unable to prevent *mode collapse*. The current paper makes two contributions. (1) It proposes a novel test for estimating support size using the *birthday paradox* of discrete probability. Using this evidence is presented that well-known GANs approaches do learn distributions of fairly low support. (2) It theoretically studies encoder-decoder GANs architectures (e.g., BiGAN/ALI), which were proposed to learn more meaningful features via GANs and (consequently) to also solve the mode-collapse issue. Our result shows that such encoder-decoder training objectives also cannot guarantee learning of the full distribution because they cannot prevent serious mode collapse. More seriously, they cannot prevent learning meaningless codes for data, contrary to usual intuition.

## 1  INTRODUCTION

From the earliest papers on Generative Adversarial Networks the question has been raised whether or not they actually come close to learning the distribution they are trained with (henceforth refered to as the target distribution)? These methods train a generator deep net that converts a random seed into a realistic-looking image. Concurrently they train a discriminator deep net to discriminate between its output and real images, which in turn is used to produce gradient feedback to improve the generator net. In practice the generator deep net starts producing realistic outputs by the end, and the objective approaches its optimal value. But does this mean the deep net has learnt the target distribution of real images? Standard analysis introduced in Goodfellow et al. (2014) shows that given "sufficiently large" generator and discriminator, sample size, and computation time the training does succeed in learning the underlying distribution arbitrarily closely (measured in Jensen-Shannon divergence). But this does not settle the question of what happens with realistic sample and net sizes.

Note that GANs differ from many previous methods for learning distributions in that they do not provide an estimate of a measure of distributional fit —e.g., perplexity score. Therefore researchers have probed their performance using surrogate qualitative tests, which were usually designed to rule out the most obvious failure mode of the training, namely, that the GAN has simply memorized

the training data. One test checks the similarity of each generated image to the nearest images in the training set. Another takes two random seeds $s_1, s_2$ that produced realistic images and checks the images produced using seeds lying on the line joining $s_1, s_2$. If such "interpolating" images are reasonable and original as well, then this may be taken as evidence that the generated distribution has many novel images. Yet other tests check for existence of semantically meaningful directions in the latent space, meaning that varying the seed along these directions leads to predictable changes e.g., (in case of images of human faces) changes in facial hair, or pose. A recent test proposed by Wu et al. (2016) checks the log-likelihoods of GANs using Annealed Importance Sampling, whose results indicate the mismatch between generator's distribution and the target distribution. Poole et al. (2016) proposed a method to trade-off between sample quality and sample diversity but they don't provide a clear definition or a quantitative metric of sample diversity.

Recently a new theoretical analysis of GANs with finite sample sizes and finite discriminator size Arora et al. (2017) revealed the possibility that training objective can sometimes approach optimality even if the generator is far from having actually learnt the distribution. Specifically, if the discriminator has size $p$, then the training objective could be $\epsilon$ close to optimal even though the output distribution is supported on only $O(p \log p/\epsilon^2)$ images. By contrast one imagines that the target distribution usually must have very large support. For example, the set of all possible images of human faces (a frequent setting in GANs work) must involve all combinations of hair color/style, facial features, complexion, expression, pose, lighting, race, etc., and thus the possible set of images of faces approaches infinity. The above paper raises the possibility that the discriminator may be unable to meaningfully distinguish such a diverse target distribution from a trained distribution with fairly small support. Furthermore, the paper notes that this failure mode is different from the one usually feared, namely the generator memorizing training samples. The analysis of Arora et al. (2017) raises the possibility that the trained distribution has small support, and yet all its samples could be completely disjoint from the training samples.

However, the above analysis was only a theoretical one, exhibiting a particular near-equilibrium solution that can happen from certain hyper-parameter combinations. It left open the possibility that real-life GANs training avoids such solutions thanks to some not-as-yet-understood property of SGD, or hyper-parameter choices. Thus further experimental investigation seems necessary. And yet it seems difficult at first sight to do such an empirical evaluation of the support size of a distribution: it is not humanly possible to go through hundreds of thousands of images, whereas automated tests of image similarity can be thrown off by small changes in lighting, pose etc.

The current paper makes two important contributions. On the empirical side it introduces a new test for the support size of the trained distribution, and uses it to find that unfortunately these mode collapse problems do arise in many well-regarded GAN training methods. On the theoretical side we prove the limitations of encoder-decoder frameworks like BiGAN (Donahue et al., 2017) and Adversarially Learned Inference or ALI (Dumoulin et al., 2017), which, inspired by autoencoder models, require the setup to learn an inference mechanism as well as a generative mechanism. The result of Arora et al. (2017) applies only to standard GAN training objectives (including JS and Wasserstein), but not to encoder-decoder setups. The clear hope in defining encoder-decoder setups is that the encoding mechanism "inverts" the generator and thus forces the generator to learn meaningful featurizations of data that are useful in downstream applications. In fact it has often been proposed that this need to learn meaningful featurizations will also solve the mode collapse problem: (Dumoulin et al., 2017) provide experiments on 2-dimensional mixtures of Gaussians suggesting this phenomenon. Our analysis shows not only that encoder-decoder training objectives cannot avoid mode collapse, but that they also cannot enforce learning of meaningful codes/features.

## 1.1 BIRTHDAY PARADOX TEST FOR SUPPORT SIZE

Let's consider a simple test that estimates the support size of a *discrete* distribution. Suppose a distribution has support $N$. The famous *birthday paradox*[1] says that a batch of about $\sqrt{N}$ samples would be likely to have a duplicate. Thus our proposed birthday paradox test for GANs is as follows.

---

[1]The following is the reason for this name. Suppose there are $k$ people in a room. How large must $k$ be before we have a high likelihood of having two people with the same birthday? Clearly, if we want 100% probability, then $k > 366$ suffices. But assuming people's birthdays are iid draws from some distribution on $[1, 366]$ it can be checked that the probability exceeds 50% even when $k$ is as small as 23.

---
**Birthday Paradox Test**

    (a) Sample a batch of $s$ images from the generator.

    (b) Use an automated measure of image similarity to flag the $\sim 20$ most similar pairs in the batch.

    (c) Visually inspect the flagged pairs and check for duplicates.

    (d) Repeat.

---

If this test reveals that batches of size $s$ have duplicate images with good probability, then suspect that the distribution has support size about $s^2$. Note that the test is not definitive, because the distribution could assign a probability $10\%$ to a single image, and be uniform on a huge number of other images. Then the test would be likely to find a duplicate even with 20 samples, though the true support size is huge. But such nonuniformity (a lot of probability being assigned to a few images) is the only failure mode of the birthday paradox test calculation, and such nonuniformity would itself be considered a failure mode of GANs training. This is captured in the following theorems:

**Theorem 1.** *Given a discrete probability distribution $P$ on a set $\Omega$, if there exists a subset $S \subseteq \Omega$ of size $N$ such that $\sum_{s \in S} P(s) \geq \rho$, then the probability of encountering at least one collision among $M$ i.i.d. samples from $P$ is $\geq 1 - \exp(-\frac{(M^2 - M)\rho}{2N})$*

**Theorem 2.** *Given a discrete probability distribution $P$ on a set $\Omega$, if the probability of encountering at least one collision among $M$ i.i.d. samples from $P$ is $\gamma$, then for $\rho = 1 - o(1)$, there exists a subset $S \subseteq \Omega$ such that $\sum_{s \in S} P(s) \geq \rho$ with size $\leq \frac{2M\rho^2}{(-3 + \sqrt{9 + \frac{24}{M}\ln\frac{1}{1-\gamma}}) - 2M(1-\rho)^2}$, under realistic assumptions on parameters.*

The proofs of these theorems are included in Appendix A. It is important to note that Theorem 1 and 2 do *not* assume that the tested distribution is uniform. In fact, Theorem 2 clarifies that if one can consistently see collisions in batches, then the distribution has a major component distribution that has *limited* support size but is almost *indistinguishable* from the full distribution via sampling a small number of samples. Thus the distribution *effectively* has small support size, which is what one should care about when sampling from it. Furthermore, without any further assumption, to accurately estimate the support size of an arbitrary distribution that has $n$ modes, $\Omega(n/\log n)$ samples need to be seen, rendering it practically infeasible for a human examiner (Valiant & Valiant, 2011).

## 2   Birthday paradox test for GANs: Experimental details

In the GAN setting, the distribution is continuous, not discrete. When support size is infinite then in a finite sample, we should not expect exact duplicate images where every pixel is identical. Thus *a priori* one imagines the birthday paradox test to completely not work. But surprisingly, it still works if we look for near-duplicates. Given a finite sample, we select the 20 closest pairs according to some heuristic metric, thus obtaining a candidate pool of potential near-duplicates inspect. Then we visually identify if any of them would be considered duplicates by humans. Our test were done using two datasets, CelebA (faces) (Liu et al., 2015) and CIFAR-10 (Krizhevsky, 2009) . Note that CelebA reasonably balanced, since the constructors intentionally made it unbiased (it contains ten thousand identities, each of which has twenty images). Also, we report in Appendix B results on the Bedroom dataset from the LSUN (Yu et al., 2015).

For faces, we find Euclidean distance in pixel space works well as a heuristic similarity measure, probably because the samples are centered and aligned. For CIFAR-10, we pre-train a discriminative CNN for the full classification problem, and use the top layer representation as an embedding. Heuristic similarity is then measured as the Euclidean distance in the embedding space. These metrics can be crude, but note that improving them can only *lower* our estimate of the support size, since a better similarity measure can only increase the number of duplicates found. Thus our reported estimates should be considered as upper bounds on the support size of the distribution.

*Note:* Some GANs (and also older methods such as variational autoencoders) implicitly or explicitly apply noise to the training and generated images. This seems useful if the goal is to compute a perplexity score, which involves the model being able to assign a nonzero probability to *every* image.

Such noised images are usually very blurry and the birthday paradox test does not work well for them, primarily because the automated measure of similarity no longer works well. Even visually judging similarity of noised images is difficult. Thus our experiments work best with GANs that generate sharper, realistic images.

## 2.1 RESULTS ON CELEBA DATASET

We test the following methods, doing the birthday paradox test with Euclidean distance in pixel space as the heuristic similarity measure For judging whether two images are the same, we set the criterion that the two faces are not exactly identical but look like doppelgangers. (Of course, in real life the only doppelgangers we know are usually twins.)

- DCGAN —unconditional, with JSD objective as described in Goodfellow et al. (2014) and Radford et al. (2015).
- MIX+ GAN protocol introduced in Arora et al. (2017), specifically, MIX+DCGAN with 3 mixture components.
- ALI (Dumoulin et al., 2017) (or equivalently BiGANs (Donahue et al., 2017)).[2]

For fair comparison, we set the discriminator of ALI (or BiGANs) to be roughly the same in size as that of the DCGAN model, since the results of Section 2.1.1 below suggests that the discriminator size has a strong effect on diversity of the learnt distribution.

We find that with probability $\geq 50\%$, a batch of $\sim 800$ samples contains at least one pair of duplicates for both DCGAN and MIX+DCGAN. Figure 1 displays duplicates and their nearest neighbors in training set. These results suggest that the support size of the distribution is less than $800^2 \approx 640000$, being at the same order of the size of the training set, but this distribution is not just memorizing the training set (see the dashed boxes).

ALI (or BiGANs) appears to be more diverse, in that collisions appear with $50\%$ probability only with a batch size of $1200$, implying a support size of a million. This is 6x the training set, but still much smaller than the diversity one would expect among human faces[3]. Nevertheless, these tests do support the suggestion in Dumoulin et al. (2017) and Donahue et al. (2017) that the bidirectional structure prevents some of the mode collapse observed in usual GANs.

### 2.1.1 DIVERSITY VS DISCRIMINATOR SIZE

The analysis of Arora et al Arora et al. (2017) suggested that the support size could be as low as near-linear in the capacity of the discriminator; in other words, there is a near-equilibrium in which a distribution of such a small support could suffice to fool the best discriminator. So it is worth investigating whether training in real life allows generator nets to exploit this "loophole" in the training that we now know is in principle available to them.

While a comprehensive test is beyond the scope of this paper, we do a first test with a simplistic version of discriminator size (i.e., capacity). We build DCGANs with increasingly larger discriminators while fixing the other hyper-parameters. The discriminator used here is a 5-layer Convolutional Neural Network such that the number of output channels of each layer is $1\times, 2\times, 4\times, 8\times dim$ where $dim$ is chosen to be $16, 24, \ldots, 120, 128$. Thus the discriminator size should be proportional to $dim^2$. Figure 2 suggests that in this simple setup the diversity of the learnt distribution does indeed grow near-linearly with the discriminator size.

## 2.2 RESULTS FOR CIFAR-10

On CIFAR-10, Euclidean distance in pixel space is not informative. So we adopt a classifying CNN with 3 convolutional layers, 2 fully-connected layer and a 10-class soft-max output pretrained with a multi-class classification objective, and use its top layer features as embeddings for similarity test

---

[2]ALI is probabilistic version of BiGANs, but their architectures are equivalent. So we only tested ALI in our experiments.

[3]After all most of us know several thousand people, but the only doppelgangers among our acquaintances are twins.

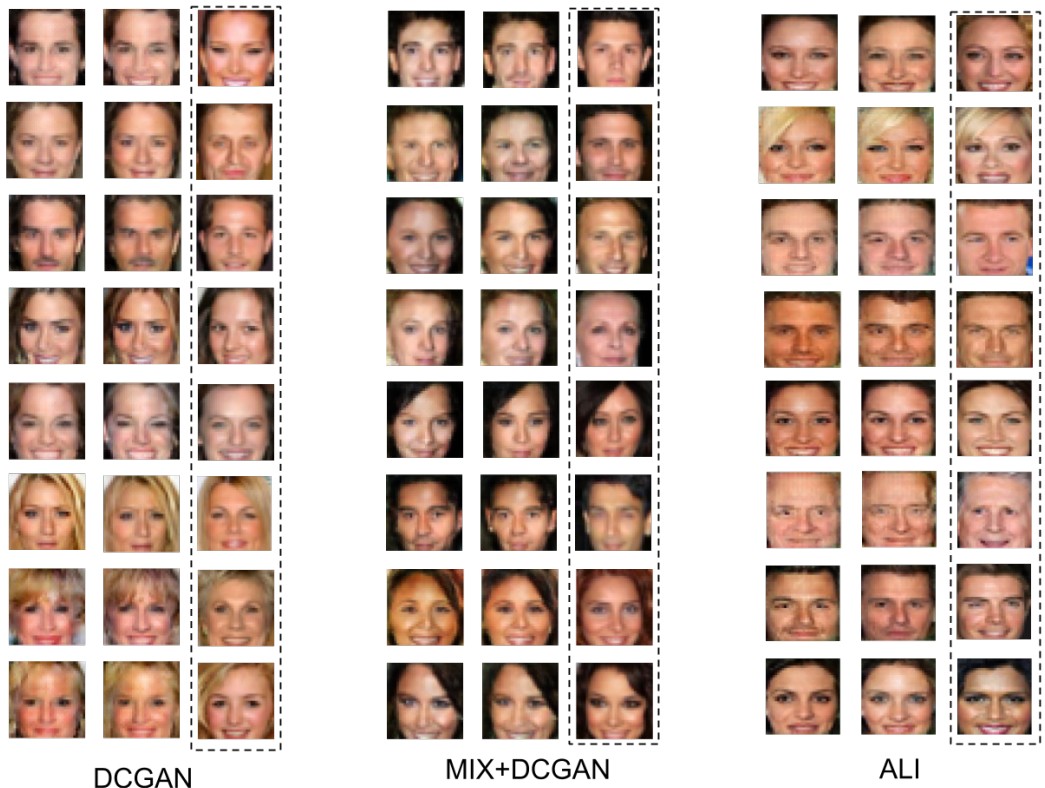

DCGAN                MIX+DCGAN                ALI

Figure 1: Most similar pairs found in batches of 800 generated faces samples from a DCGAN, a MIX+DCGAN (with 3 component) and an ALI. Each pair is from a different batch. Shown in dashed boxes are the nearest neighbors in training data.

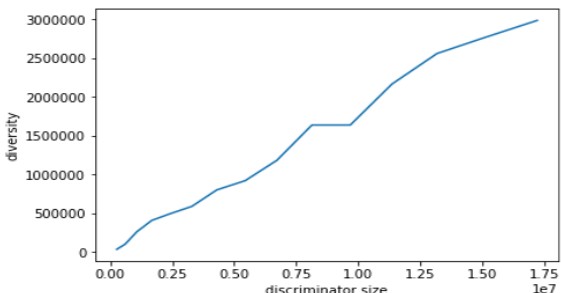

Figure 2: Diversity's dependence on discriminator size. The diversity is measured as the square of the batch size needed to encounter collision w.p. $\geq 50\%$ v.s. size of discriminator.

using Euclidean distance. Firstly, we find that the result of the test is affected by the quality of samples. If the training uses noised samples (with noise being added either explicitly or implicitly in the objective) then the generated samples are also quite noisy. Then the most similar samples in a batch tend to be blurry blobs of low quality. Indeed, when we test a DCGAN (even the best variant with 7.16 Inception Score reported in Huang et al. (2016)), the pairs returned are mostly blobs.

To get meaningful test results, we turn to a Stacked GAN which is the state-of-the-art generative model on CIFAR-10 (Inception Score 8.59 (Huang et al., 2016)). It also generates the most real-looking images. Since this model is trained by conditioning on class label, we measure its diversity within each class separately. The batch sizes needed for duplicates are shown in Table 1. Duplicate samples shown in Figure 3.

| Aeroplane | Auto-Mobile | Bird | Cat | Deer | Dog | Frog | Horse | Ship | Truck |
|-----------|-------------|------|-----|------|-----|------|-------|------|-------|
| 500 | 50 | 500 | 100 | 500 | 300 | 50 | 200 | 500 | 100 |

Table 1: Class specific batch size needed to encounter duplicate samples with $> 50\%$ probability, from a Stacked GAN trained on CIFAR-10

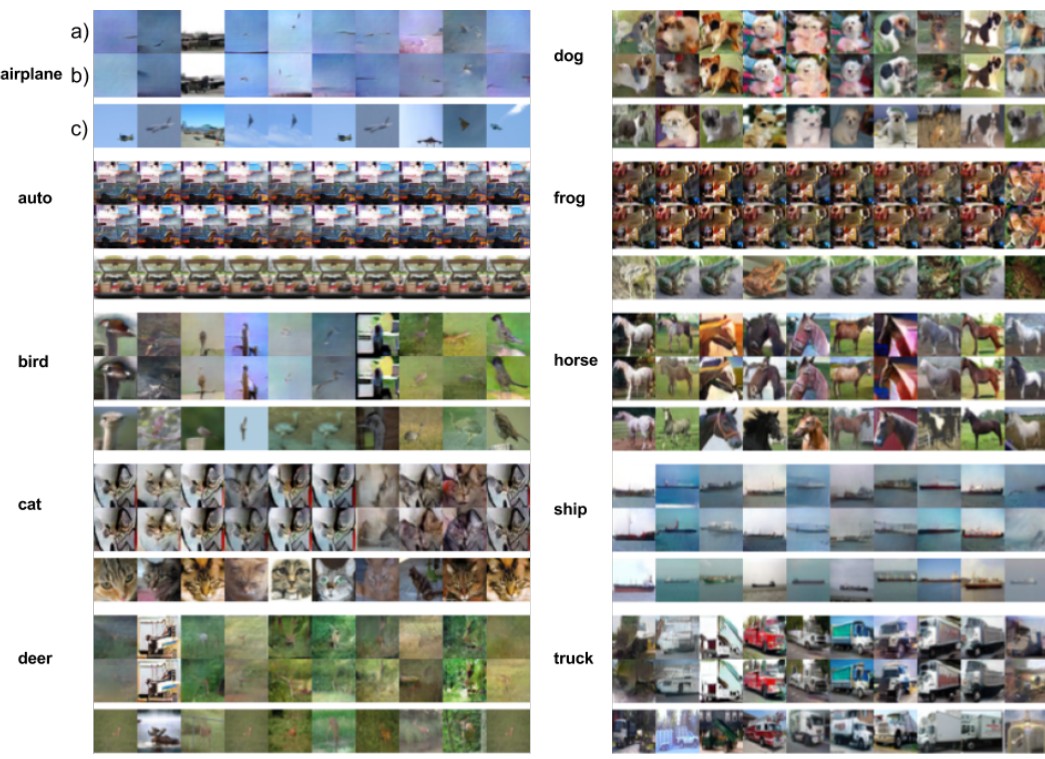

Figure 3: Duplicate pairs found in a batch of 1000 generated CIFAR-10 samples from a Stacked GAN.a)-b):pairs of duplicate samples; c): nearest neighbor of b) in the training set.

We check whether the detected duplicates are close to any of the training images, by looking for the nearest neighbor in the training set using our heuristic similarity measure and visually inspecting the closest suspects. We find that the closest image is quite different from the duplicates, suggesting the issue with GANs is indeed lack of diversity (low support size) instead of memorizing training set.

## 3 LIMITATIONS OF ENCODER-DECODER GAN ARCHITECTURES

We recall the Adversarial Feature Learning (BiGAN) setup from Donahue et al. (2017). The "generative" player consists of two parts: a generator $G$ and an encoder $E$. The generator takes as input a latent variable $z$ and produces a sample $G(z)$; the encoder takes as input a data sample $x$ and produces a guess for the latent variable $E(x)$. This produces two joint distributions over pairs of latent variables and data samples: $(z, G(z))$ and $(E(x), x)$. The goal of the "generative" player is to convince the discriminator that these two distributions are the same, whereas the discriminator is being trained to distinguish between them. In the ideal case, the hope is that the "generative" player converges to $(z, G(z))$ and $(E(x), x)$ both being jointly distributed as $p(z, x)$ where $p$ is the joint distribution of the latent variables and data – i.e. $G(z)$ is distributed as $p(x|z)$: the true generator distribution; and $E(x)$ is distributed as $p(z|x)$: the true encoder distribution.

Using usual min-max formalism for adversarial training, the BiGAN objective is written as:

$$\min_{G,E} \max_{D} \left| \underset{x \sim \hat{\mu}}{\mathbb{E}} \phi(D(x, E(x))) - \underset{z \sim \hat{\nu}}{\mathbb{E}} \phi(D(G(z), z)) \right| \tag{1}$$

where $\hat{\mu}$ is the empirical distribution over data samples $x$; $\hat{\nu}$ is a distribution over random "seeds" for the latent variables: typically sampled from a simple distribution like a standard Gaussian; and $\phi$ is a concave "measuring" function. (The standard choice is $\log$, though other options have been proposed in the literature.) For our purposes, we will assume that $\phi$ outputs values in the range $[-\Delta, \Delta], \Delta \geq 1$, and is $L_\phi$-Lipschitz.

For ease of exposition we will refer to $\mu$ as the *image distribution*. The proof is more elegant if we assume that $\mu$ consists of images that have been noised —concretely, think of replacing every 100th pixel by Gaussian noise. Such noised images would of course look fine to our eyes, and we would expect the learnt encoder/decoder to not be affected by this noise. For concreteness, we will take the seed/code distribution $\nu$ to be a spherical zero-mean Gaussian (in an arbitrary dimension and with an arbitrary variance). [4]

Furthermore, we will assume that $\text{Domain}(\mu) = \mathbb{R}^d$, $\text{Domain}(\nu) = \mathbb{R}^{\tilde{d}}$ with $\tilde{d} < d$ (we think of $\tilde{d} \ll d$, which is certainly the case in practice). As in Arora et al. (2017) we assume that discriminators are $L$-lipschitz with respect to their trainable parameters, and the support size of the generator's distribution will depend upon this $L$ and the capacity $p$ (= number of parameters) of the discriminator.

**Theorem 3** (Main). *There exists a generator $G$ of support $\frac{p\Delta^2 \log^2(p\Delta LL_\phi/\epsilon)}{\epsilon^2}$ and an encoder $E$ with at most $\tilde{d}$ non-zero weights, s.t. for all discriminators $D$ that are $L$-Lipschitz and have capacity less than $p$, it holds that*

$$|\mathop{\mathbb{E}}_{x\sim\mu} \phi(D(x, E(x))) - \mathop{\mathbb{E}}_{z\sim\nu} \phi(D(G(z), z))| \leq \epsilon$$

The interpretation of the above theorem is as stated before: the encoder $E$ has very small complexity (we will subsequently specify it precisely and show it simply extracts noise from the input $x$); the generator $G$ is a small-support distribution (so presumably far from the true data distribution). Nevertheless, the value of the BiGAN objective is small. The argument of Arora et al. (2017) seems unable to apply to this setting. It is a simple concentration/epsilon-net argument showing that the discriminator of capacity $p$ cannot distinguish between a generator that samples from $\mu$ versus one that memorizes a subset of $\frac{p \log p}{\epsilon^2}$ random images in $\mu$ and outputs one randomly from this subset. By contrast, in the current setting we need to say what happens with the encoder.

*The precise noise model:* Denoting by $\tilde{\mu}$ the distribution of unnnoised images, and $\nu$ the distribution of seeds/codes, we define the distribution of noised images $\mu$ as the following distribution: to produce a sample in $\mu$ take a sample $\tilde{x}$ from $\tilde{\mu}$ and $z$ from $\nu$ independently and output $x = \tilde{x} \circledast z$, which is defined as

$$x_i = \begin{cases} z_{\frac{i}{\lfloor \frac{d}{\tilde{d}} \rfloor}}, & \text{if } i \equiv 0 (\text{ mod } \lfloor \frac{d}{\tilde{d}} \rfloor) \\ \tilde{x}_i, & \text{otherwise} \end{cases}$$

In other words, set every $\lfloor \frac{d}{\tilde{d}} \rfloor$-th to one of the coordinates of $z$. In practical settings, $\tilde{d} \ll d$ is usually chosen, so noising $\tilde{d}$ coordinates out of all $d$ will have no visually noticeable effect. [5]

### 3.1 PROOF SKETCH, THEOREM 3; DETAILED PROOF IN APPENDIX

The main idea is to show the existence of the generator/encoder pair via a probabilistic construction that is shown to succeed with high probability.

- *Encoder $E$*: The encoder just extracts the noise from the noised image (by selecting the relevant $\tilde{d}$ coordinates). Namely, $E(\tilde{x} \circledast z) = z$. (So the *code* is just gaussian noise and has no meaningful content.) It's easy to see this can be captured using a ReLU network with $\tilde{d}$ weights: we can simply connect the $i$-th output to the $(i\lfloor \frac{d}{\tilde{d}} \rfloor)$-th input using an edge of weight 1.

---

[4] The proof can be extended to non-noised inputs, by assuming that natural images have an innate stochasticity that can be extracted and to more general code distributions.

[5] Note the result itself doesn't require constraints on $d, \tilde{d}$ beyond $\tilde{d} < d$ – the point we are stressing is merely that the noise model makes sense for practical choices of $\tilde{d}, d$.

- *Generator $G$:* This is designed to produce a distribution of support size $m :=$ $\frac{p\Delta^2 \log^2(p\Delta LL_\phi/\epsilon)}{\epsilon^2}$. We first define a partition of $\text{Domain}(\nu) = \mathbb{R}^{\tilde{d}}$ into $m$ equal-measure blocks under $\nu$. Next, we sample $m$ samples $x_1^*, x_2^*, \ldots, x_m^*$ from the image distribution. Finally, for a sample $z$, we define $\text{ind}(z)$ to be the index of the block in the partition in which $z$ lies, and define the generator as $G(z) = x_{\text{ind}(z)}^* \circledast z$. Since the set of samples $x_i^* : i \in [m]$ is random, this specifies a *distribution* over generators. We prove that with high probability, one of these generators satisfies the statement of Theorem 3. Moreover, we show that such a generator can be easily implemented using a ReLU network of complexity $O(md)$ in Theorem 5.

The basic intuition of the proof is as follows. We will call a set $T$ of samples from $\nu$ *non-colliding* if no two lie in the same block. Let $\mathcal{T}_{nc}$ be the distribution over non-colliding sets $\{z_1, z_2, \ldots, z_m\}$, s.t. each $z_i$ is sampled independently from the conditional distribution of $\nu$ inside the i-th block.

First, we notice that under the distribution for $G$ we defined, it holds that

$$\mathbb{E}_{x \sim \mu} \phi(D(x, E(x))) = \mathbb{E}_{x \sim \tilde{\mu}, z \sim \nu} \phi(D(x \circledast z, z)) = \mathbb{E}_G \mathbb{E}_{z \sim \nu} \phi(G(z), z)$$

In other words, the "expected" encoder correctly matches the expectation of $\phi(D(x, E(x)))$, so that the discriminator is fooled. We want to show that $\mathbb{E}_G \mathbb{E}_{z \sim \nu} \phi(G(z), z)$ concentrates enough around this expectation, as a function of the randomness in $G$, so that we can say with high probability over the choice of $G$, $|\mathbb{E}_{x \sim \mu} \phi(D(x, E(x))) - \mathbb{E}_{z \sim \nu} \phi(G(z), z)|$ is small. We handle the concentration argument in two steps:

First, we note that we can calculate the expectation of $\phi(D(G(z), z))$ when $z \sim \nu$ by calculating the empirical expectation over $m$-sized non-colliding sets $T$ sampled according to $\mathcal{T}_{nc}$. Namely, as we show in Lemma D.1:

$$\mathbb{E}_{z \sim \nu} \phi(D(G(z), z)) = \mathbb{E}_{T \sim \mathcal{T}_{nc}} \mathbb{E}_{z \sim T} \phi(D(G(z), z))$$

Thus, we have reduced our task to arguing about the concentration of $\mathbb{E}_{T \sim \mathcal{T}_{nc}} \mathbb{E}_{z \sim T} \phi(D(G(z), z))$ (viewed as a random variable in $G$). Towards this, we consider the random variable $\mathbb{E}_{z \sim T} \phi(D(G(z), z))$ as a function of the randomness in $G$ and $T$ both. Since $T$ is a non-colliding set of samples, we can write

$$\mathbb{E}_{z \sim T} \phi(D(G(z), z)) = f(x_i^*, z_i, i \in [m])$$

for some function $f$, where the random variables $x_i^*$, $z_i$ are all mutually independent – thus use McDiarmid's inequality to argue about the concentration of $f$ in terms of both $T$ and $G$.

From this, we can use Markov's inequality to argue that all but an exponentially small (in $p$) fraction of encoders $G$ satisfy that: for all but an exponentially small (in $p$) fraction of non-colliding sets $T$, $|\mathbb{E}_{z \sim T} \phi(D(G(z), z)) - \mathbb{E}_G \mathbb{E}_{T \sim \mathcal{T}_{nc}} \mathbb{E}_{z \sim T} \phi(D(G(z), z))|$ is small. Note that this has to hold *for all* discriminators $D$ – so we need to additionally build an epsilon-net, and union bound over all discriminators, similarly as in Arora et al. (2017). Then, it's easy to extrapolate that for such $G$,

$$|\mathbb{E}_{T \sim \mathcal{T}_{nc}} \mathbb{E}_{z \sim T} \phi(D(G(z), z)) - \mathbb{E}_G \mathbb{E}_{T \sim \mathcal{T}_{nc}} \mathbb{E}_{z \sim T} \phi(D(G(z), z))|$$

is small, as we want. The details are in Lemma D.2 in the Appendix.

## 4 CONCLUSIONS

The paper reveals gaps in current thinking about GANs, and hopes to stimulate further theoretical and empirical study. GANs research has always struggled with the issue of mode collapse, and recent theoretical analysis of Arora et al. (2017) shows that the GANs training objective is not capable of preventing mode collapse. This exhibits the *existence* of bad solutions in the optimization landscape. This in itself is not definitive, since existence of bad solutions is also known for the more traditional classification tasks Zhang et al. (2017), where heldout sets can nevertheless prove that a good solution has been reached. The difference in case of GANs is lack of an obvious way to establish that training succeeded.

Our new Birthday Paradox test gives a new benchmark for testing the support size (i.e., diversity of images) in a distribution. Though it may appear weak, experiments using this test suggest that current GANs approaches, specifically, the ones that produce images of higher visual quality, do suffer mode collapse. Our rough experiments also suggest —again in line with the previous theoretical analysis—that the size of the distribution's support scales near-linearly with discriminator capacity.

Researchers have raised the possibility that the best use of GANs is not distribution learning but feature learning. Encoder-decoder GAN architectures seem promising since they try to force the generator to use "meaningful" encodings of the image. While such architectures do exhibit slightly better diversity in our experiments, our theoretical result suggest that the the encoder-decoder objective is also unable to avoid mode collapse, furthermore, also fails to guarantee meaningful codes.

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

## A  BIRTHDAY PARADOX CALCULATION

In this section, we prove the statemts of Theorems 1 and 2.

*Proof of Theorem 1.*

$$\Pr[\text{there is at least a collision among } M \text{ samples}]$$
$$\geq 1 - \Pr[\text{there is no collision within set } S \text{ among } M \text{ samples}]$$
$$\geq 1 - 1 \times (1 - \frac{\rho}{N}) \times (1 - \frac{2\rho}{N}) \times \cdots \times (1 - \frac{(M-1)\rho}{N})$$
$$\geq 1 - \exp(-\frac{(M^2 - M)\rho}{2N}) \tag{2}$$

We use the fact that the worst case is when the $\rho$ probability mass is uniformly distributed on $S$.

$\square$

*Proof of Theorem 2.* Suppose $X_1, X_2, \ldots$ are i.i.d. samples from the discrete distribution $P$. We define $T = \inf\{t \geq 2, X_t \in \{X_1, X_2, \ldots, X_{t-1}\}\}$ to be the collision time and also we use $\beta = \frac{1}{\Pr[T=2]} = \frac{1}{\sum_{X \in \Omega} P(X)^2}$ as a surrogate for the uniformity of $P$. According Theorem 3 in Wiener (2005), $Pr[T \geq M]$ can be upper-bounded using $\beta$. Specifically, with $\beta > 1000$ and $M \leq 2\sqrt{\beta \ln \beta}$, which is usually true when $P$ is the distribution of a generative model of images,

$$\Pr[T \geq M] \geq \exp(-\frac{M^2}{2\beta} - \frac{M^3}{6\beta^2})$$

To estimate $\beta$, we notice that

$$\Pr[T \geq M] = 1 - \Pr[T \leq M] = 1 - \Pr[\text{there is at least a collision among } M] = 1 - \gamma \geq \exp(-\frac{M^2}{2\beta} - \frac{M^3}{6\beta^2})$$

which immediately implies

$$\beta \leq \frac{2M}{-3 + \sqrt{9 + \frac{24}{M} \ln \frac{1}{1-\gamma}}} = \beta^*$$

This gives us a upper-bound of the uniformity of distribution $P$, which we can utilize. Let $S \subseteq \Omega$ be the smallest set with probability mass $\geq \rho$ and suppose it size is $N$. To estimate the largest possible $N$ such that the previous inequality holds, we let

$$\frac{1}{(\frac{\rho}{N})^2 N + (1-\rho)^2} \leq \beta^*$$

from which we obtain

$$N \leq \frac{2M\rho^2}{(-3 + \sqrt{9 + \frac{24}{M} \ln \frac{1}{1-\gamma}}) - 2M(1-\rho)^2}$$

$\square$

## B  EXPLORATORY RESULTS ON BEDROOM DATASET

We test DCGANs trained on the trained on $64 \times 64$ center cropped Bedroom dataset(LSUN) using Euclidean distance to extract collision candidates since it is impossible to train a CNN classifier on such single-category (bedroom) dataset. We notice that the most similar pairs are likely to be the corrupted samples with the same noise pattern (top-5 collision candidates all contain such patterns). When ignoring the noisy pairs, the most similar "clean" pairs are not even similar according to human eyes. This implies that the distribution puts significant probability on noise patterns, which can be seen as a form of under-fitting (also reported in the DCGAN paper). We manually counted

the number of samples with a fixed noise pattern from a batch of 900 i.i.d samples. We find 43 such corrupted samples among the 900 generated images, which implies $43/900 \approx 5\%$ probability.

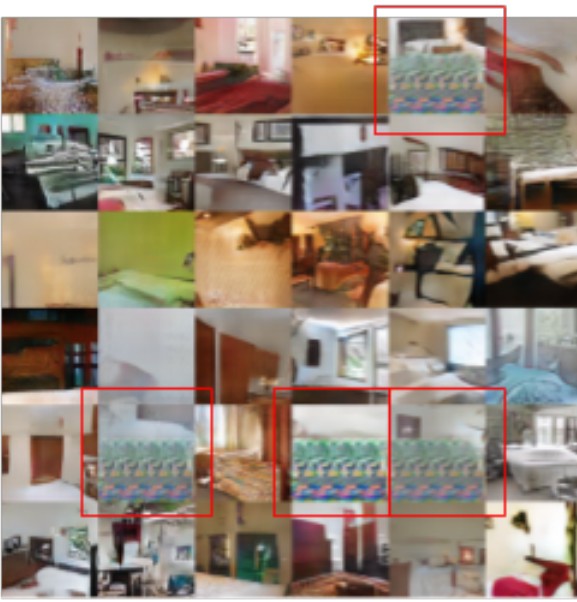

Figure 4: Randomly generated samples from a DCGAN trained on Bedroom dataset. Note that there are corrupted images with a fixed noise pattern (emphasized in red boxes).

## C  BIRTHDAY PARADOX TEST FOR VAES

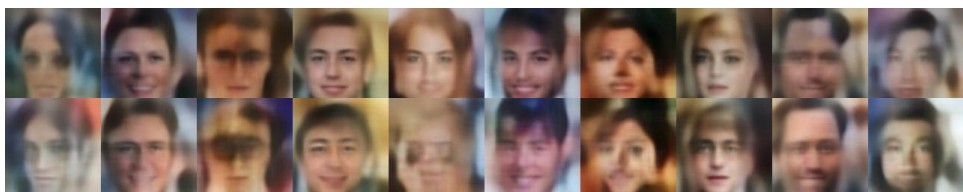

Figure 5: Collision candidates found in Variation Auto-Encoder samples. The duplicated samples are frequently blurry ones because the crucial features (eyes, hair, mouth) of VAE samples are not as distinctive as those of GANs'.

Given these findings, it is natural to wonder about the diversity of distributions learned using earlier methods such as Variational Auto-Encoders Kingma & Welling (2014) (VAEs). Instead of using feedback from the discriminator, these methods train the generator net using feedback from an approximate perplexity calculation. Thus the analysis of Arora et al. (2017) does not apply as is to such methods and it is conceivable they exhibit higher diversity. However, we find the birthday paradox test difficult to run since samples from a VAE trained on CelebA are not realistic or sharp enough for a human to definitively conclude whether or not two images are almost the same. Fig 5 shows examples of collision candidates found in batches of 400 samples; clearly some indicative parts (hair, eyes, mouth, etc.) are quite blurry in VAE samples.

# D  BiGAN'S DON'T EVADE THE CURSE OF LOW DIVERSITY

We recall the basic notation from the main part: the image distribution will be denoted as $\mu$, and the code/seed distribution as $\nu$, which we assume is a spherical Gaussian. For concreteness, we assumed the domain of $\tilde{\mu}$ is $\mathbb{R}^d$ and the domain of $\nu$ is $\mathbb{R}^{\tilde{d}}$ with $\tilde{d} \leq d$. (As we said, we are thinking of $\tilde{d} \ll d$.)

We also introduced the quantity $m := \frac{p\Delta^2 \log^2(p\Delta LL_\phi/\epsilon)}{\epsilon^2}$.

Before proving Theorem 3, let's note that the claim can easily be made into a finite-sample version. Namely:

**Corollary D.1** (Main, finite sample version). *There exists a generator $G$ of support $m$, s.t. if $\hat{\mu}$ is the uniform distribution over a training set $S$ of size at least $m$, and $\hat{\nu}$ is the uniform distribution over a sample $T$ from $\nu$ of size at least $m$, for all discriminators $D$ that are $L$-Lipschitz and have less than $p$ parameters, with probability $1 - \exp(-\Omega(p))$ over the choice of training set $S,T$ we have:*

$$| \underset{x \sim \hat{\mu}}{\mathbb{E}} \phi(D(x, E(x))) - \underset{z \sim \hat{\nu}}{\mathbb{E}} \phi(D(G(z), z))| \leq \epsilon$$

*Proof.* As is noted in Theorem B.2 in Arora et al. (2017), we can build a $\frac{\epsilon}{LL_\phi}$-net for the discriminators with a size bounded by $e^{p \log(LL_\phi p/\epsilon)}$. By Chernoff and union bounding over the points in the $\frac{\epsilon}{LL_\phi}$-net, with probability at least $1 - \exp(-\Omega(p))$ over the choice of a training set $S$, we have

$$| \underset{x \sim \mu}{\mathbb{E}} \phi(D(x, E(x))) - \underset{x \sim \hat{\mu}}{\mathbb{E}} \phi(D(x, E(x)))| \leq \frac{\epsilon}{2}$$

for all discriminators $D$ with capacity at most $p$. Similarly, with probability at least $1 - \exp(-\Omega(p))$ over the choice of a noise set $T$,

$$| \underset{z \sim \nu}{\mathbb{E}} \phi(D(G(z), z)) - \underset{z \sim \hat{\nu}}{\mathbb{E}} \phi(D(G(z), z))| \leq \frac{\epsilon}{2}$$

Union bounding over these two events, we get the statement of the Corollary.

$\square$

Spelling out the distribution over generators more explicitly, we will in fact show:

**Theorem 4** (Main, more detailed). *Let $G$ follow the distribution over generators defined in Section 3. With probability $1 - \exp(-\Omega(p \log(\Delta/\epsilon)))$ over the choice of $G$, for all discriminators $D$ that are $L$-Lipschitz and have capacity bounded by $p$:*

$$| \underset{x \sim \mu}{\mathbb{E}} \phi(D(x, E(x))) - \underset{z \sim \nu}{\mathbb{E}} \phi(D(G(z), z))| \leq \epsilon$$

## D.1  PROOF OF THE MAIN CLAIM

Let us recall we call a set $T$ of samples from $\nu$ *non-colliding* if no two lie in the same block and we denoted $\mathcal{T}_{nc}$ to be the distribution over non-colliding sets $\{z_1, z_2, \ldots, z_m\}$, s.t. each $z_i$ is sampled independently from the conditional distribution of $\nu$ inside the i-th block of the partition.

First, notice the following Lemma:

**Lemma D.1** (Reducing to expectations over non-colliding sets). *Let $G$ be a fixed generator, and $D$ a fixed discriminator. Then,*

$$\underset{z \sim \nu}{\mathbb{E}} \phi(D(G(z), z)) = \underset{T \sim \mathcal{T}_{nc}}{\mathbb{E}} \underset{z \sim T}{\mathbb{E}} \phi(D(G(z), z))$$

*Proof.* By definition,

$$\underset{T \sim \mathcal{T}_{nc}}{\mathbb{E}} \underset{z \sim T}{\mathbb{E}} \phi(D(G(z), z)) = \frac{1}{m} \sum_{i=1}^{m} \underset{z_i \sim (\mathcal{T}_{nc})_i}{\mathbb{E}} \phi(D(G(z_i), z_i))$$

where $(\mathcal{T}_{nc})_i$ is the conditional distribution of $\nu$ in the $i$-th block of the partition. However, since the blocks form an equipartitioning, we have

$$\frac{1}{m}\sum_{i=1}^{m}\mathop{\mathbb{E}}_{z_i\sim(\mathcal{T}_{nc})_i}\phi(D(G(z_i),z_i)) = \sum_{i=1}^{m}\mathop{\Pr}_{z\sim\nu}(z\text{ belongs to cell }i)\mathop{\mathbb{E}}_{z_i\sim(\mathcal{T}_{nc})_i}\phi(D(G(z_i),z_i))$$
$$= \mathop{\mathbb{E}}_{z\sim\nu}\phi(D(G(z),z))$$

$\square$

**Lemma D.2** (Concentration of good generators). *With probability* $1-\exp(-\Omega(p\log(\Delta/\epsilon)))$ *over the choice of* $G$,

$$|\mathop{\mathbb{E}}_{T\sim\mathcal{T}_{nc}}\mathop{\mathbb{E}}_{z\sim T}\phi(D(G(z),z)) - \mathop{\mathbb{E}}_{G}\mathop{\mathbb{E}}_{T\sim\mathcal{T}_{nc}}\mathop{\mathbb{E}}_{z\sim T}\phi(D(G(z),z))| \le \epsilon$$

*for all discriminators $D$ of capacity at most $p$.*

*Proof.* Consider $\mathbb{E}_{z\sim T}\phi(D(G(z),z))$ as a random variable in $T\sim\mathcal{T}_{nc}$ and $G$ for a fixed $D$. We can write

$$\mathop{\mathbb{E}}_{z\sim T}\phi(D(G(z),z)) = f(x_i^*, z_i, i\in[m])$$

where the random variables $x_i^*, z_i$ are all mutually independent. Note that the arguments that $f$ is a function of are all independent variables, so we can apply McDiarmid's inequality. Towards that, let's denote by $z_{-i}$ the vector of all inputs to $f$, except for $z_i$. Notice that

$$|f(z_{-i},z_i) - f(z_{-i},\tilde{z}_i)| \le \frac{1}{m}, \forall i\in[m] \tag{3}$$

(as changing $z_i$ to $\tilde{z}_i$ only affect one out of the $m$ terms in $E_{z\sim T}\phi(D(G(z),z))$). Analogously we have

$$|f(x_{-i}^*,x_i^*) - f(x_{-i}^*,\tilde{x}_i^*)| \le \frac{1}{m}, \forall i\in[m] \tag{4}$$

Denoting $R_{D,T,G} = f(x_i^*,z_i,i\in[m]) - \mathbb{E}_{T,G}[f(x_i^*,z_i,i\in[m])]$, by McDiarmid we get

$$\mathop{\Pr}_{T,G}(|R_{D,T,G}|\ge\epsilon) \le \exp(-\Omega(m\epsilon^2))$$

Building a $\frac{\epsilon}{LL_\phi}$-net for the discriminators and union bounding over the points in the net, we get that $\Pr_{T,G}(\exists D, |R_{D,T,G}|\ge\epsilon/2) \le \exp(-\Omega(p\log(\Delta/\epsilon)))$. On the other hand, we also have

$$\mathop{\Pr}_{T,G}(\exists D, |R_{D,T,G}|\ge\epsilon/2) = \mathop{\mathbb{E}}_{G}[\mathop{\Pr}_{T}(\exists D, |R_{D,T,G}|\ge\epsilon)]$$

so by Markov's inequality, with probability $1-\exp(-\Omega(p\log(\Delta/\epsilon))$ over the choice of $G$, ,

$$\mathop{\Pr}_{T}(\exists D, |R_{D,T,G}|\ge\epsilon/2)] \le \exp(-\Omega(p\log(\Delta/\epsilon)))$$

Let us denote by $\tilde{T}(G)$ the sets $T$, s.t. $\forall D, |R_{D,T,G}| \le \epsilon/2$. Then, with probability $1-\exp(-\Omega(p\log(\Delta/\epsilon))))$ over the choice of $G$, we have that for all $D$ of capacity at most $p$,

$$|\mathop{\mathbb{E}}_{T}\mathop{\mathbb{E}}_{z\sim T}\phi(D(G(z),z)) - \mathop{\mathbb{E}}_{G,T}\mathop{\mathbb{E}}_{z\sim T}\phi(D(G(z),z))|$$
$$\le \int_{T\in\tilde{T}(G_x)} |\mathop{\mathbb{E}}_{z\sim T}\phi(D(G(z),z)) - \mathop{\mathbb{E}}_{G,T}\mathop{\mathbb{E}}_{z\sim T}\phi(D(G(z),z))|d\mathcal{T}_{nc}(T)$$
$$+ \int_{T\notin\tilde{T}(G_x)} |\mathop{\mathbb{E}}_{z\sim T}\phi(D(G(z),z)) - \mathop{\mathbb{E}}_{G,T}\mathop{\mathbb{E}}_{z\sim T}\phi(D(G(z),z))|d\mathcal{T}_{nc}(T)$$
$$\le \epsilon/2 + \exp(-\Omega(p\log(\Delta/\epsilon)))$$
$$\le \epsilon/2 + O((\epsilon/\Delta)^p)$$
$$\le \epsilon$$

which is what we want.

$\square$

With these in mind, we can prove the main theorem:

*Proof of Theorem 3.* From Lemma D.1 and Lemma D.2 we get that with probability $1 - \exp(-\Omega(p\log(\Delta/\epsilon)))$ over the choice of $G$,

$$\left| \mathbb{E}_{z\sim\nu} \phi(D(G(z),z)) - \mathbb{E}_{G,T\sim\mathcal{T}_G} \phi(D(G(z),z)) \right| \leq \epsilon$$

for all discriminators $D$ of capacity at most $p$. On the other hand, it's easy to see that

$$\mathbb{E}_{G,T\sim\mathcal{T}_G} \phi(D(G(z),z)) = \mathbb{E}_{x\sim\tilde{\mu},z\sim\nu} \phi(D(x \circledast z, z))$$

Furthermore,

$$\mathbb{E}_{x\sim\mu} \phi(D(x, E(x))) = \mathbb{E}_{x\sim\tilde{\mu},z\sim\nu} \phi(D(x \circledast z, z))$$

by the definition of $E(x)$, which proves the statement of the theorem.

$\square$

## D.2 Representability results

In this section, we show that a generator $G$ of the type we described in the previous section can be implemented easily using a ReLU network. The encoder can be parametrized by the set $x_1^*, x_2^*, \ldots, x_m^*$ of "memorized" training samples. The high-level overview of the network is rather obvious: we partition $\mathbb{R}^{\tilde{d}}$ into equal-measure blocks; subsequently, for a noise sample $z$, we determine which block it belongs to, and output the corresponding sample $x_i^*$, which is memorized in the weights of the network.

**Theorem 5.** *Let $G$ be the generator determined by the samples $x_1^*, x_2^*, \ldots, x_m^* \in \mathbb{R}^d$, i.e. $G(z) = x_{ind(z)}^* \circledast z$. For any $\delta > 0$, there exists a ReLU network with $O(md)$ non-zero weights, which implements a function $\tilde{G}$, s.t. $\|\tilde{G}(\nu) - G(\nu)\|_{TV} \leq \delta$, where $\|\cdot\|_{TV}$ denotes total variation distance.*[6]
[7]

*Proof.* The construction of the network is pictorially depicted on Figure D.2. We expand on each of the individual parts of the network.

First, we show how to implement the partitioning into sectors. The easiest way to do this is to use the fact that the coordinates of a spherical Gaussian are independent, and simply partition each dimension separately into equimeasure blocks, depending on the value of $|z_i|$: the absolute value of the $i$-th coordinate. Concretely, without loss of generality, let's assume $m = k^{\tilde{d}}$, for some $k \in \mathbb{N}$. Let us denote by $\tau_i \in \mathbb{R} : i \in [k]$, $\tau_0 = 0$ the real-numbers, s.t. $\Pr_{z_i\sim\nu_i}[|z_i| \in (\tau_{i-1}, \tau_i)] = \frac{1}{k}$. . We will associate to each $d$-tuple $(i_1, i_2, \ldots, i_d) \in [k]^d$ the cell $\{z \in \mathbb{R}^{\tilde{d}} : |z_j| \in (\tau_{i_j-1}, \tau_{i_j}), \forall j \in [\tilde{d}]\}$. These blocks clearly equipartition $\mathbb{R}^{\tilde{d}}$ with respect to the Gaussian measure.

Inferring the $\tilde{d}$-tuple after calculating the absolute values $|z_j|$ (which is trivially representable as a ReLU network as $\max(0, z_j) + \max(0, -z_j)$) can be done using the "selector" circuit introduced in 'Arora et al. (2017). Namely, by Lemma 3 there, there exists a ReLU network with $O(k)$ non-zero weights that takes $|z_j|$ as input and outputs $k$ numbers $(b_{j,1}, b_{j,2}, \ldots, b_{j,k})$, s.t. $b_{j,i_j} = 1$ and $b_{j,l} = 0, j \neq i_j$ with probability $1 - \delta$ over the randomness of $z_j, j \in [\tilde{d}]$.

Since we care about $G(\nu)$ and $\tilde{G}(\nu)$ being close in total variation distance only, we can focus on the case where all $z_j$ are such that $b_{j,i_j} = 1$ and $b_{j,l} = 0, j \neq i_j$ for some indices $i_j$.

We wish to now "turn on" the memorized weights for the corresponding block in the partition. To do this, we first pass the calculated $\tilde{d}$-tuple through a network which interprets it as a number in

---

[6] The notation $G(\nu)$ simply denotes the distribution of $G(z)$, when $z \sim \nu$.

[7] For smaller $\delta$, the weights get larger, so the bit-length to represent them gets larger. This is standard when representing step functions using ReLU, for instance see Lemma 3 in Arora et al. (2017). For the purposes of our main result Theorem 3, it suffices to make $\delta = O(\frac{\epsilon}{\Delta})$, which translates into a weights on the order of $O(\frac{m\epsilon}{\Delta})$ – which in turn translates into bit complexity of $O(\log(m\epsilon/\Delta))$ so this isn't an issue as well.

$k$-ary and calculates it's equivalent decimal representation. This is easily implementable as a ReLU network with $O(\tilde{d}k)$ weights calculating $L = \sum_{j=1}^{\tilde{d}} k^j \sum_{l=1}^{k} b_{j,l} i_j$. Then, we use use a simple circuit with $O(m)$ non-zero weights to output $m$ numbers $(B_1, B_2, \ldots, B_m)$, s.t. $B_L = 1$ and $B_i = 0, i \neq L$ (implemented in the obvious way). The subnetwork of $B_i$ will be responsible for the $i$-th memorized sample.

Namely, we attach to each coordinate $B_i$ a curcuit with fan-out of degree $d$, s.t. the weight of edge $j \in [d]$ is $x_{i,j}^*$. Let's denote these outputs as $F_{i,j}, i \in [m], j \in [d]$ and let $\tilde{F}_i : i \in [d]$ be defined as $\tilde{F}_i = \sum_{j=1}^{m} F_{i,j}$. It's easy to see since $B_i = 0, i \neq L$ that $\tilde{F}_i = x_{L,i}^*$.

Finally, the operation $\circledast$ can be trivially implemented using additional $d$ weights: we simply connect each output $i$ either to $z_{\frac{i}{\lfloor \frac{d}{d} \rfloor}}$ if $i \equiv 0 \pmod{\lfloor \frac{d}{d} \rfloor}$ or to $\tilde{F}_i$ otherwise.

Adding up the sizes of the individual components, we see the total number of non-zero weights is $O(md)$, as we wanted.

$\square$

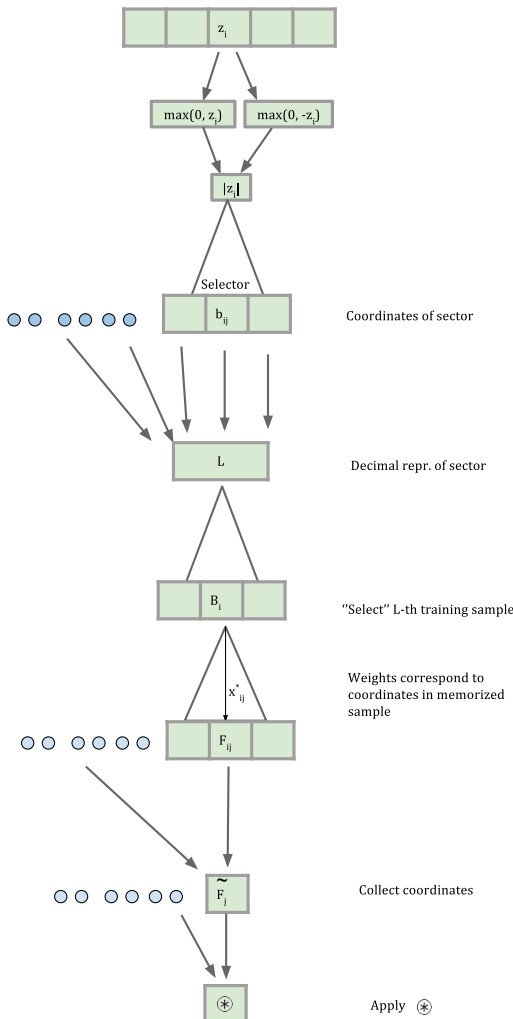

Figure 6: A ReLU implementation of a memorizing generator

