# OpenReview forum: "Do GANs learn the distribution? Some Theory and Empirics"
_ICLR.cc/2018/Conference — Accept (Poster)_

### Official Review · AnonReviewer3 · 2017-11-27
**Do GANs learn the distribution? Some Theory and Empirics**

**Rating:** 7
**Confidence:** 4

**Review:**

The paper adds to the discussion on the question whether Generative Adversarial Nets (GANs) learn the target distribution. Recent theoretical analysis of GANs by Arora et al. show that of the discriminator capacity of is bounded, then there is a solution the closely meets the objective but the output distribution has a small support. The paper attempts to estimate the size of the support for solutions produced by typical GANs experimentally. The main idea used to estimate the support is the Birthday theorem that says that with probability at least 1/2, a uniform sample (with replacement) of size S from a set of  N elements will have a duplicate given S > \sqrt{N}. The suggested plan is to manually check for duplicates in a sample of size s and if duplicate exists, then estimate the size of the support to be s^2. One should note that the birthday theorem assumes uniform sampling.  In the revised versions, it has been clarified that the tested distribution is not assumed to be uniform but the distribution has "effectively" small support size using an indistinguishability notion. Given this method to estimate the size of the support, the paper also tries to study the behaviour of estimated support size with the discriminator capacity. Arora et al. showed that the output support size has nearly linear dependence on the discriminator capacity. Experiments are conducted in this paper to study this behaviour by varying the discriminator capacity and then estimating the support size using the idea described above. A result similar to that of Arora et al. is also given for the special case of Encoder-Decoder GAN.

Evaluation:
Significance: The question whether GANs learn the target distribution is important and any  significant contribution to this discussion is of value.

Clarity: The paper is written well and the issues raised are well motivated and proper background is given.

Originality: The main idea of trying to estimate the size of the support using a few samples by using birthday theorem seems new.

Quality: The main idea of this work is to give a estimation technique for the support size for the output distribution of GANs.

---

> ### Author Response · Authors · 2017-12-14
> **Response to AnonReviewer3**
>
> It is important to note that Theorem 1 and 2 do *not* assume that the tested distribution is uniform. (The birthday paradox holds even if human birthdays are distributed in a highly nonuniform way.) This confusion possibly underlies the reviewer’s score.
>
> Theorem 2 clarifies that if one can consistently see collisions in batches, then the distribution has a major component that has *limited* support size but is almost *indistinguishable* from the full distribution via sampling a small number of samples. (For example, it could assign very tiny probability to a lot of other images.) Thus the distribution *effectively* has small support size, which is what one should care about when sampled from.  We will try other phrasing of that section to clarify this issue further.
>
> It may help to point out (as proven in paper [1] below) that to correctly estimate support size of a distribution with n modes, at least  n / log n samples need to be seen by the human examiner. Since support size is ~10^6 in some GANs studied here, examining n / log n is infeasible for a human. Though conceivably some follow-up work could do this via a giant mechanical turk experiment. We will be sure to add these notes to the final version so other readers are not confused.
> (Possibly the reviewer is also alluding to the possibility that CelebA dataset is a highly nonuniform distribution of faces. This is possible, but the constructors [2] tried hard to make it unbiased (it contains ten thousand identities, each of which has twenty images) [2]. Also, we report results on it because it was used in many GANs papers.)
>
> To the best of our knowledge, our birthday paradox test ---which is of course related to classical ideas in statistics--- is more rigorous and quantitative than past tests for mode collapse we are aware of.
>
> Finally, the reviewer appears to have missed the important theoretical contribution showing how encoder-decoder GANs may learn un-informative codes.
>
> [1]  Valiant, Gregory, and Paul Valiant, Estimating the Unseen: An n/log(n)-sample Estimator for Entropy and Support Size, Shown Optimal via New CLTs, STOC 2011
> [2] Liu, Ziwei, Ping Luo, Xiaogang Wang, and Xiaoou Tang. Deep Learning Face Attributes in the Wild

---

### Official Review · AnonReviewer1 · 2017-11-27
**Extremely interesting topic; insightful but limited method and theory**

**Rating:** 6
**Confidence:** 4

**Review:**

This paper proposes a clever new test based on the birthday paradox for measuring diversity in generated samples. The main goal is to quantify mode collapse in state-of-the-art generative models. The authors also provide a specific theoretical construction that shows bidirectional GANs cannot escape specific cases of mode collapse.
Using the birthday paradox test, the experiments show that GANs can learn and consistently reproduce the same examples, which are not necessarily exactly the same as training data (eg. the triplets in Figure 1).
The results are interpreted to mean that mode collapse is strong in a number of state-of-the-art generative models.
Bidirectional models (ALI, BiGANs) however demonstrate significantly higher diversity that DCGANs and MIX+DCGANs.
Finally, the authors verify empirically the hypothesis that diversity grows linearly with the size of the discriminator.

This is a very interesting area and exciting work. The main idea behind the proposed test is very insightful. The main theoretical contribution stimulates and motivates much needed further research in the area. In my opinion both contributions suffer from some significant limitations. However, given how little we know about the behavior of modern generative models, it is a good step in the right direction.


1. The biggest issue with the proposed test is that it conflates mode collapse with non-uniformity. The authors do mention this issue, but do not put much effort into evaluating its implications in practice, or parsing Theorems 1 and 2. My current understanding is that, in practice, when the birthday paradox test gives a collision I have no way of knowing whether it happened because my data distribution is modal, or because my generative model has bad diversity. Anecdotally, real-life distributions are far from uniform, so this should be a common issue. I would still use the test as a part of a suite of measurements, but I would not solely rely on it. I feel that the authors should give a more prominent disclaimer to potential users of the test.

2. Also, given how mode collapse is the main concern, it seems to me that a discussion on coverage is missing. The proposed test is a measure of diversity, not coverage, so it does not discriminate between a generator that produces all of its samples near some mode and another that draws samples from all modes of the true data distribution. As long as they yield collisions at the same rate, these two generative models are ‘equally diverse’. Isn’t coverage of equal importance?

3. The other main contribution of the paper is Theorem 3, which shows—via a very particular construction on the generator and encoder—that bidirectional GANs can also suffer from serious mode collapse. I welcome and are grateful for any theory in the area. This theorem might very well capture the underlying behavior of bidirectional GANs, however, being constructive, it guarantees nothing in practice. In light of this, the statement in the introduction that “encoder-decoder training objectives cannot avoid mode collapse” might need to be qualified. In particular, the current statement seems to obfuscate the understanding that training such an objective would typically not result into the construction of Theorem 3.

---

> ### Author Response · Authors · 2017-12-14
> **Response to AnonReviewer1**
>
> Thank you for your positive and detailed comment! We’ll address your concerns point by point.
>
> “Conflates mode collapse with non-uniformity” “coverage”
> It is important to clarify ---though very likely the reviewer understood--- that Theorem 1 and 2 hold without the assumption of uniformity. (The birthday paradox holds even though human birthdays are not uniformly distributed.)  That said, the reviewer is correct that our test does not test for coverage and we will add a disclaimer to this effect. We note that testing coverage of n in general requires at least n/log n samples. (See our response to 3rd reviewer.)
>
> Indeed, we are assuming that CelebA dataset is reasonably well-balanced (it contains ten thousand identities, each of which has twenty images) [2], and therefore a GAN that produces a highly non-uniform distribution of faces is some kind of failure mode. It is conceivable that CelebA is not well-constructed for the reasons mentioned by the reviewer, but it has been used in most previous GANs papers, so it was natural to report our findings on that. In the final version we’ll put a suitable disclaimer about this issue.
>
>
> “Practical implication of Theorem 3?”
> The reviewer is correct that we have only shown *existence* of a bad equilibrium, not proved that SGD or other algorithms *find* it. (Analysing SGD’s behavior for deep learning is of course an open problem.) But note that some of the problems raised by Theorem 3 are observed in practice too; see e.g. empirical studies ([1], [2]) which suggest that BiGANs/ALI can learn un-informative codes. We’ll rewrite to make these issues clearer.
>
> [1] Chunyuan Li, Hao Liu, Changyou Chen, Yunchen Pu, Liqun Chen, Ricardo Henao and Lawrence Carin, ALICE: Towards Understanding Adversarial Learning for Joint Distribution Matching, NIPS 2017
> [2] Jun-Yan Zhu*    Taesung Park*    Phillip Isola    Alexei A. Efros Unpaired image-to-image translation using cycle-consistent adversarial networks. ICCV, 2017.

---

### Official Review · AnonReviewer2 · 2017-11-30
**The paper uses birthday paradox to show experimentally that some most popular GAN architectures generate distributions with fairly low support. Also some theoretical explanation for the phenomena is given**

**Rating:** 7
**Confidence:** 3

**Review:**

The article "Do GANs Learn the Distribution? Some Theory and Empirics" considers the important problem of quantifying whether the distributions obtained from generative adversarial networks come close to the actual distribution of images. The authors argue that GANs in fact generate the distributions with fairly low support.

The proposed approach relies on so-called birthday paradox which allows to estimate the number of objects in the support by counting number of matching (or very similar) pairs in the generated sample. This test is expected to experimentally support the previous theoretical analysis by Arora et al. (2017). The further theoretical analysis is also performed showing that for encoder-decoder GAN architectures the distributions with low support can be very close to the optimum of the specific (BiGAN) objective.

The experimental part of the paper considers the CelebA and CIFAR-10 datasets. We definitely see many very similar images in fairly small sample generated. So, the general claim is supported. However, if you look closely at some pictures, you can see that they are very different though reported as similar. For example, some deer or truck pictures. That's why I would recommend to reevaluate the results visually, which may lead to some change in the number of near duplicates and consequently the final support estimates.

To sum up, I think that the general idea looks very natural and the results are supportive. On theoretical side, the results seem fair (though I didn't check the proofs) and, being partly based on the previous results of Arora et al. (2017), clearly make a step further.

---

> ### Author Response · Authors · 2017-12-14
> **Response to AnonReviewer2**
>
> Thank you for the positive and careful review! We agree that a judgement call needs to be made for assessing whether two images are “essentially” the same. For the final version of the paper we will utilize a second human examiner and report collisions only if both examiners judge the image to be same. It is correct that this may slightly affect the estimate of support size, though we expect the conclusions to not change too much.

---

### Author Response · Authors · 2018-01-06
**Thanks for the reviews and comments!**

We've made a few minor revisions to the manuscript, mostly for clarity and brevity.

---

### Decision · Program_Chairs · 2018-01-29
**ICLR 2018 Conference Acceptance Decision**

**Decision:**

Accept (Poster)

**Comment:**

* presents a novel way analyzing GANs using the birthday paradox and provides a theoretical construction that shows bidirectional GANs cannot escape specific cases of mode collapse
* significant contribution to the discussion of whether GANs learn the target disctibution
* thorough justifications